# Measuring electric-acoustic heterodyning in piezoelectric materials
Tomasz Karpisz [1,2], Robert L. Lirette [1,2], Aaron M. Hagerstrom[1], Nathan D. Orloff[1] & Angela C. Stelson [1] ✉

Many electrically active devices rely on nonlinear signal mixing (heterodyning) between two electrical signals. Heterodyning between electric and acoustic signals can allow for active control of typically passive components such as transmission lines, acoustic resonators, and electrical resonators built from piezoelectric materials. However, there are few techniques to characterize the nonlinear properties of materials that lead to heterodyning between electric and acoustic signals within the material. Here we demonstrate a proof-of-concept microwave interferometer that uses electromagnetic and acoustic waves to measure second-order mixing from electrical and acoustic signals in a piezoelectric material. The sum and difference frequencies of signal mixing can be detected in the electromagnetic spectrum in our measurement. We show the effect of frequency and power of the fundamental signals on the mixing products. We additionally characterize the heterodyne signal to show that it is electric-acoustic in nature, versus purely electric. Characterizing nonlinear electric-acoustic properties is important to the development of next generation piezoelectric materials models and devices.

Nonlinear mixing of electromagnetic signals can be understood as two-signal multiplication. In a quadratic nonlinear mixing process, new signals are generated at the sum and difference frequencies of two original signals. To the best of our knowledge, these electrical nonlinear processes were first described in the context of vacuum tubes[1,2] but today are also used in diodes[3,4], transistors[5,6] and photonic crystal fibers[7,8]. Nonlinear mixing of acoustic signals has also been used to up- and down-convert electrical signals on-chip[9]. Applications of nonlinear processes that lead to two-wave mixing are typically between different frequencies of the same domain-either electric or acoustic. However, recent reports demonstrate the promise of nonlinear mixing between the electric and acoustic domains in applications ranging from quantum computing to biosensing[10]. These applications rely on nonlinear materials physics beyond the piezoelectric effect to create electric-acoustic heterodyned signals.

Electric-acoustic mixing has been used to achieve dynamic phase modulation in on-chip surface acoustic wave devices at microwave frequencies for signal control at cryogenic temperatures[10]. The mechanism to achieve phase modulation relies on the electric-acoustic materials properties (second order piezoelectric coefficient) of lithium niobate ($LiNbO_3$) to generate the electric-acoustic mixing products in the acoustic domain.

The combination of ultrasound with electrical sensing can be used in biomedical applications for detection of small signals from deep within human tissue that can be used for imaging purposes[11]. In this case, an ultrasound signal and an electrical signal were applied to a biofluid phantom, and the resulting mixing product is measured. The nonlinear effect here is driven by the change in conductivity of the solution as a function of pressure. While these examples illustrate the promise of leveraging nonlinear electric-acoustic materials properties, metrology techniques to quantify these materials properties are lacking. The biggest challenge to measuring nonlinear electric-acoustic materials properties is that the signals associated with these effects are small, and high power density of both electrical and acoustic signals is typically required[12]. Additionally, electric-acoustic properties are expected to vary as a function of both electric and acoustic frequency, as well as crystal orientation. A robust measurement for these properties should be highly sensitive, extensible to a range of frequencies and power levels, and allow for a range of sample orientations and geometries.

In this work, we developed a method for measuring nonlinear electric-acoustic mixing with high sensitivity. We demonstrate the proof-of-concept of this measurement technique with piezoelectric lead zirconium titanate (PZT) and show electric-acoustic mixing products at approximately 6 GHz. This response is due to the electrostrictive effect, a second order nonlinear materials property. Inspired by nonlinear electrical measurements[12] and interferometer measurements of the piezoelectric coefficient[13], we developed an interferometer-based approach to measure nonlinear electric-acoustic signals. Nonlinear electric-acoustic materials properties beyond the

[1]National Institute of Standards and Technology, Boulder, CO, USA. [2]University of Colorado, Boulder, CO, USA. ✉e-mail: angela.stelson@nist.gov

piezoelectric effect are typically small and difficult to detect. An interferometer-based approach allows us to apply large electrical signals to our sample and cancel the fundamental signal. This strategy improves signal-to-noise when detecting the mixing product and allows detection of signals that would be masked by noise. The interferometer approach developed here is advantageous compared to other nonlinear electric-acoustic techniques in the literature[12] because it is more sensitive due to the reduction of noise floor with the interferometer. The electric-acoustic mixing product in our interferometer is generated in a PZT sample that is stimulated by an ultrasound transducer and placed on coplanar waveguide (CPW). We characterize the response of the PZT as a function of acoustic and electric power levels and validate the transduction between electrical and acoustic domains. Our measurement approach is generalizable to enable characterization of electric-acoustic mixing in materials for electric-acoustic heterodyning applications. The approach we develop here is flexible with low-cost instruments and can be readily scaled to higher electrical powers.

## Results
### Theory
To devise a general method to measure second-order nonlinear mixing, we developed a microwave interferometer. Typically, interferometers are based on splitting a signal into two identical paths and matching phases and losses of those arms at a certain frequency. Any changes to the electrical length of the system can be detected as a change of relative transmission amplitude or phase. Changes in electrical length can be due to either changes in attenuation or phase delay of one of the interferometer arms. Interferometer-style measurements typically have lower noise-floor levels over a frequency range that provides a larger dynamic range for measuring small signals such as electric-acoustic mixing products.

Before we describe the measurement setup used here, it is worthwhile to explore how interferometers operate and how wave cancellation can be achieved. Let us consider two identical electrical signal paths. The signal has frequency $f_1$ and amplitude $E_0$. In this case, the paths have identical loss $\alpha_1$ and phase constant $\beta_1 = 2\pi f_1 \sqrt{\mu\varepsilon}$, where $\varepsilon$ is permittivity and $\mu$ is magnetic permeability. To describe how an interferometer works at frequencies other than $f_1$, we now assume that both signal paths are propagated in a single-mode waveguide with effective loss and phase constant. For this model, at frequency $f_1 \pm \delta_f$, the phase and (loss) constant is slightly different than $\beta_1$ ($\alpha_1$) by $\delta_\beta$ ($\delta_\alpha$), while the physical length of the interferometer arms is constant. Therefore, the signal path length difference of $d_1 - d_2$ that resulted in full wave cancelation originally will no longer result in full cancellation:

$$E_0 e^{j\left((\omega_1 \pm \delta_\omega)t - (\beta_1 \pm \delta_\beta)d_1\right) - (\alpha_1 \pm \delta_\alpha)d_1} + E_0 e^{j\left((\omega_1 \pm \delta_\omega)t - (\beta_1 \pm \delta_\beta)d_2\right) - (\alpha_1 \pm \delta_\alpha)d_2}$$

$$= E_0 e^{j(\phi_1 \pm \delta_\phi) - \Delta_\alpha} + E_0 e^{j(\phi_1 \pm \delta_\phi + \pi/2) - \Delta_\alpha} \neq 0$$

(1)

where: $\delta_\omega = 2\pi\delta_f$, $\triangle_\alpha$ is loss caused by propagation of electromagnetic wave in both interferometer arms, $\delta_\phi$ is additional phase difference at $f_1 \pm \delta_f$ frequency. To estimate the cancellation in our measurement beyond the null of the interferometer and develop an expectation for our measurement, we simulated a section of the setup that contains our PZT sample. We used a 2D finite element simulation software to simulate the propagation constant (Fig. 1a) of a coplanar waveguide loaded with PZT with the same dimensions as the one used in the experiment (Fig. 1b, details in (see Supplementary Note: 1 for supporting details).

If we assume two identical 2-cm sections of PZT-loaded transmission lines and the experimental resonance frequency $f_1 \approx 5.892$ GHz, we can calculate the expected transmission as a function of frequency using Eq. (1). For the case of identical length and loss from the two sections, the transmission will be $e^{-2\alpha_1 d_1}$ (yellow dashed line in Fig. 1c). This simplified calculation illustrates the qualitative frequency dependence of our interferometer, however in practice any imbalance of the interferometer

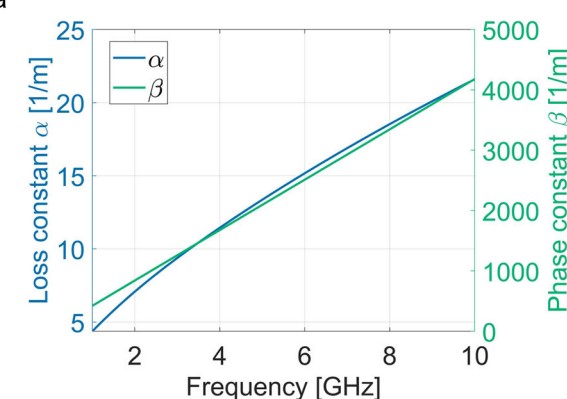

a

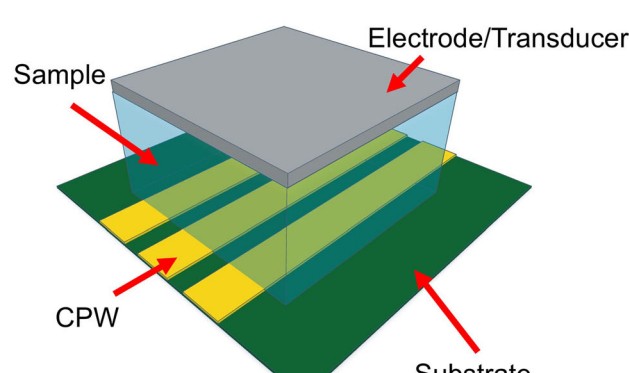

b

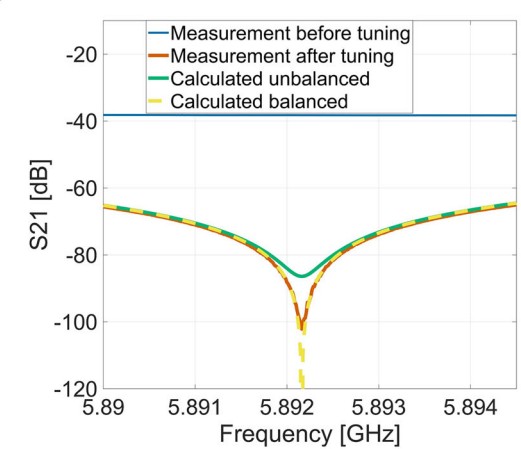

c

**Fig. 1 | Coplanar waveguide interferometer design. a** Loss ($\alpha$) and phase constant ($\beta$) of electromagnetic wave propagating in coplanar waveguide (CPW) with sample on top. **b** Model of the sample over the CPW with transducer on top. **c** Measured and calculated transmission |S21| in an interferometer. The blue line is the measured interferometer |S21| before matching attenuation on both arms. The orange line is the measured |S21| after matching losses of both arms. The green line is calculated |S21| interferometer spectrum with 0.01% smaller signal amplitude in one interferometer arm. The yellow dashed line is the calculated |S21| with balanced losses in both arms.

arm loss degrades performance. The influence of a loss mismatch of 0.01% between interferometer arms is shown on Fig. 1c with green line. To show how our interferometer performs compared to our predictions, we plot the measured transmission of an interferometer setup. The maximum cancellation is around −100 dB which is above the noise floor of the spectrum analyzer, but well below 0.01%.

**Fig. 2 | Electric-acoustic mixing product measurements. a** Picture of experimental interferometer with two coplanar waveguides (CPW) arms, one loaded with piezoelectric lead zirconium titanate (PZT) interfacing with an acoustic transducer. **b** Interferometer measurement setup schematic, where (VNA) is Vector Network Analyzer used to tune the interferometer, (HF Source) is the electrical signal source supplying signal to the interferometer during electric-acoustic measurements, and (SA) is the spectrum analyzer measuring the resulting signals. The red dashed rectangle shows interferometer components: quadrature couplers (Coupler), and two interferometer arms indicated with pink and orange dashed line boxes that contains: coplanar waveguides (CPW), a programable attenuator (Att), and the sample (PZT). The acoustic source components (blue line) consist of: an acoustic frequency source (AF source), an amplifier (AMP1), and an acoustic transducer exciting the PZT material under test (AT). The green box indicates sensing acoustic transducer path that consists of: a side-sensing transducer (ST), an amplifier (AMP2), and an oscilloscope (OSC) that is used to detect acoustic signals in the sample. **c** SA frequency sweep with acoustic transducer OFF (blue) and ON (orange dashed). **d** Difference of signal amplitude at SA frequency sweep between acoustic transducer ON and OFF.

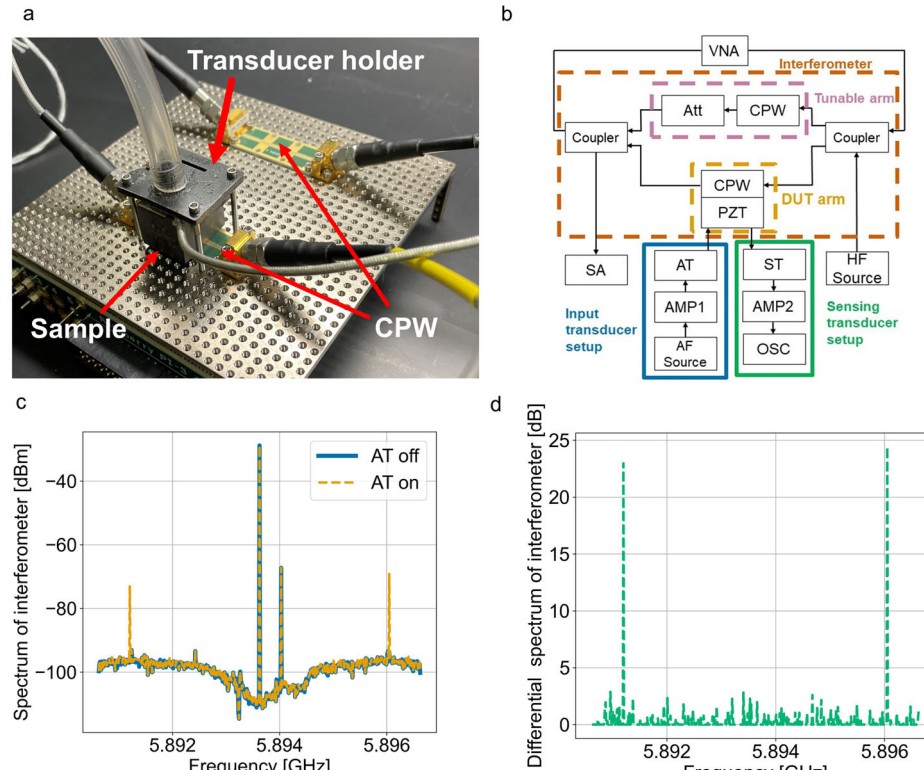

In this work we use our interferometer setup to show nonlinear mixing of two signals. The relevant frequencies are $f_{null}$ which is the frequency the interferometer is tuned to (see Fig. 1c). The second frequency ($f_a$) is provided by acoustic signal and depends on the transducer. The third frequency ($f_1$) is the frequency of the electrical signal that is routed into the CPW. The frequencies $f_{null}$ and $f_1$ may be set equal to each other but are not necessarily the same depending on the experimental configuration. Because we are measuring a piezoelectric material, there are also electromagnetic waves at $f_1$ and $f_a$. We show that nonlinear mixing product ($f_1 \pm f_a$) can be measured as both an acoustic signal and an electromagnetic signal. Generally, nonlinear effects in this setup can be associated either with linear and nonlinear electromagnetic wave interactions[14–16] or acoustic modulation of electromagnetic material properties. We focus on describing our analysis of those nonlinearities.

## Experimental setup

The electrical portion of our interferometer setup was built from connectorized components combined with modular printed circuit board components (Fig. 2a). To split and combine the electromagnetic wave, quadrature couplers divided the signal into equal outputs, each delayed by a quarter of a period relative to each other. To allow us to measure acoustic and electromagnetic wave interaction in piezoelectric materials, a section of each interferometer arm transitioned to a modular CPW circuit board. The CPW geometry allowed us to place blocks of material on top of the coplanar waveguide.

We mounted acoustic transducers on the sample to introduce an acoustic signal and allow for electric-acoustic mixing. The acoustic transducer was powered by a programmable source (AF Source), whose signal was amplified by a 30 dB amplifier (AMP1) with maximum power output of 3 W. Here we report acoustic power levels as reported by AF Source unless specified otherwise. To determine the resonant frequency of the transducer, we measured the reflection coefficient with a vector network analyzer (VNA) and found the unloaded transducer to have a minimum reflection and resonant frequency at ~2.5 MHz. To estimate the loaded resonance frequency of the transducer, we measured reflection of the AF Source signal

in the measurement setup and used the frequency with the smallest reflection value (~2.2 MHz). We used this value as our default frequency of the transducer for the experiment.

Delivering acoustic waves into the PZT sample increased the sample temperature due to the acoustic absorption in the sample. The PZT sample and acoustic transducer movement was limited by a holder that is also used to direct air flow and control the temperature of the transducer throughout the measurement (see Fig. 2a). Turning on the airflow at maximum pressure of 90 PSI while using maximum transducer power allowed us to keep the temperature below 33 °C (10 °C higher than room temperature) measured at the top of the transducer. This led to an internal PZT temperature increase, which affected the loss and phase constant of the interferometer arm. We allowed the system to thermally equilibrate for ~10 min before tuning the interferometer to ensure system electrical stability. The arm without the PZT sample block included a programmable attenuator (0.05 dB attenuation step size, Att in Fig. 2b) in its path. This attenuator allowed us to tune the electrical path losses to match the acoustically excited arm.

## Interferometer measurements

Before making measurements of electric-acoustic mixing, we used an automated procedure to tune the interferometer. A vector network analyzer (VNA) allowed us to track the transmission spectra minimum and optimize attenuator settings to minimize transmission at $f_{null}$ (the null of the interferometer). The effects of tuning can be seen on Fig. 1c where the blue line shows transmission of interferometer when the attenuator was set to its maximum level of 120 dB, giving an overall system attenuation of ~−38 dB. The orange line shows interferometer transmission after matching attenuation in both arms of the interferometer, lowering the transmission by 64 dB at $f_{null}$. In our tuning procedure, we did not tune the frequency of the interferometer null $f_{null}$. Instead, we found the electrical frequency that minimizes the transmission to determine $f_{null}$. We found that a physical length difference in the length of the interferometer arms of 30 cm resulted in transmission minimum every 500 MHz which allowed us to pick a frequency of interest with sufficient resolution. Adjusting the frequency of the

system $f_{null}$ would be possible by including a tunable phase shifter but those have slightly different attenuation for each phase shift. Adding a phase shifter would complicate the tuning process as any correction of the frequency would force changes in attenuator settings which would change the frequency again. After tuning the interferometer, the VNA source power was turned off to eliminate stray signals.

With the interferometer tuned, we apply a single tone at frequency $f_1$ with the HF source (Fig. 2b). The interferometer measurement can in principle be used in two ways with different advantages: 1) $f_1$ can be set as the fundamental frequency provided by the HF source frequency) to be $f_{null}$ (interferometer null), or 2) $f_{null}$ can be set as the mixing product frequency $f_1 \pm f_a$. Figure 2c, shows configuration 1, which cancels the incoming electrical fundamental signal from the HF source. In principle, configuration 1) is not beneficial for lower powers as the interferometer attenuation is around $-70$ dB which will not affect noise floor generated by HF source at $f_1 \pm f_a$, which is $\sim -100$ dB. Configuration 2 lowers the noise floor at $f_1 \pm f_a$ as the interferometer cancelation is $\sim -110$ dB at $f_1 \pm f_a$, which gives 10 dB larger dynamic range of measurement.

The results in Fig. 2c are presented for a nominal HF power level of 0 dBm, and a nominal AF power level of 0 dBm, as measured at the source. We observe that $f_{null}$ varies slightly between experiments and we attribute this to temperature-induced losses of the PZT sample. It is worth noting that the shape of interferometer well is different at Figs. 1c and 2c because the VNA power is turned off for the interferometer measurement, so the only signal being cancelled at most frequencies is the noise from the spectrum analyzer. In this setup, we also observe a smaller peak next to $f_{null}$ when the acoustic source is turned off, and we attribute this to nonlinearities of the HF source used in this experiment. We used configuration 1 for all measurements at higher power levels where the pre-amplifier in the spectrum analyzer was turned off. We used configuration 2 for all measurements at lower power where the pre-amplifier was turned on.

When the AF source is turned on, we observe two mixing product peaks at $f_1 \pm f_a$, where $f_1$ is the frequency of the HF source ($\sim 5.9$ GHz for all results presented here) and $f_a$ is the AF source frequency (2.5 MHz for this measurement). In Fig. 2d, we plot the difference of the two measurements, which shows the impact of the acoustic signal introduced by AT. We observe the two signals $A(f_1 \pm f_a)$ that are products of nonlinear mixing between electromagnetic and acoustic signals.

To characterize our measurement system, we tracked the amplitude of four signals in the measured spectrum: $A(f_1)$, $A(f_a)$, and $A(f_1 \pm f_a)$ (Fig. 3). First, we measured the power levels of these four signals at constant AF source and HF source power of 4.77 dBm and 9 dBm, respectively. Then, we varied the AF source frequency (Fig. 3a). As expected, the signal power at $f_1$ remained constant, since the signal was unchanged during this experiment, while the power at $f_a$ varied as a function of frequency. This is attributed primarily to the frequency dependence of the efficiency of the transducer, which we expect to vary substantially in a narrow frequency range[13]. The highest power level we observed for $f_a$ was at $\sim 2.2$ MHz. These measurements were repeated 10 times and error bars represent Type A uncertainty evaluated from repeatability of the experiments. The error bars represent one standard deviation of data values at each frequency point.

We can formulate an equation describing power amplitude of nonlinear mixing products:

$$A(f_1 \pm f_a) = A(f_1)A(f_a)Q(f_1, f_a)\mu \qquad (2)$$

where $Q$ stands for nonlinear signal mixing coefficient, $A(f_1)$ is the amplitude of the input electrical signal, $A(f_a)$ is the amplitude of the input acoustic signal, and $\mu$ is the complex attenuation of the interferometer arm section where the nonlinear signal propagates. We note that $\mu$ describes the portion of the interferometer (cables, connectors, etc.) comprised of passive components, and we expect this parameter to be constant as a function of power. The mixing coefficient $Q$ is related to the electrostrictive properties of the piezoelectric material (see Supplementary Note: 2 for supporting details)[14,15]. In this model of the material, we can expect $Q$ to vary smoothly

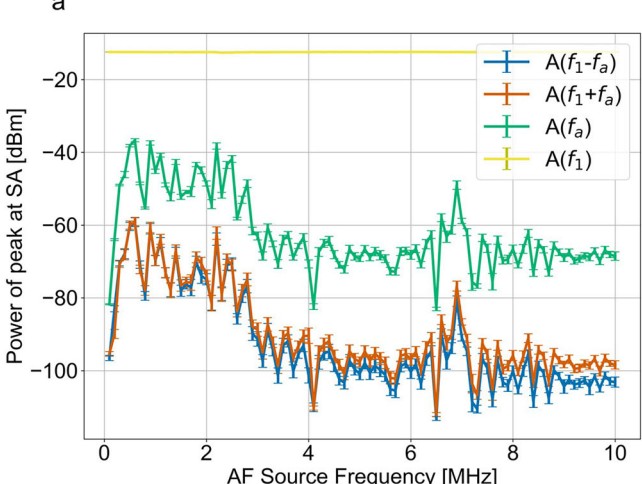

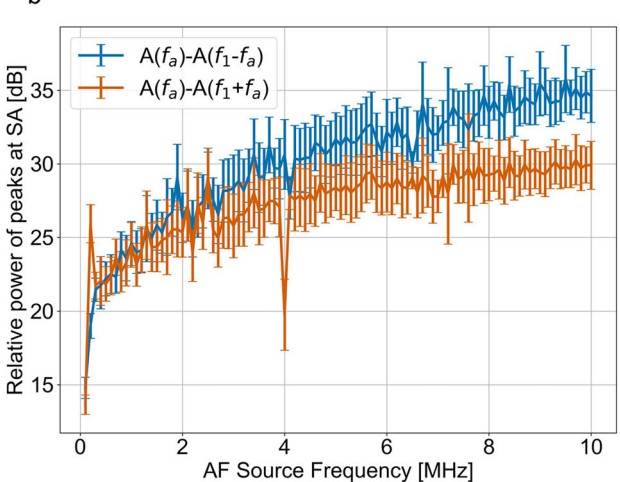

**Fig. 3 | Frequency dependence of electric-acoustic signal. a** Power of electric-acoustic product mixing peak $A(f_1 \pm f_a)$ and fundamental signal frequencies $f_1$ and $f_a$ as a function of acoustic frequency (AF) transducer source frequency. **b** Power difference of electric-acoustic product mixing peak $A(f_1 \pm f_a)$ and fundamental acoustic signal $A(f_a)$ as a function of AF transducer source frequency.

as a function of electrical and acoustic frequency. However, $Q$ should be independent of the amplitude of the electric and acoustic input signals. The electrostrictive coefficient $m_{ijkl}$ can be related to the coefficient $Q$ from Eq. (2) by creating a geometric model of the linear electrical and acoustic properties of the coplanar waveguide device[16]. These calibrations are beyond the scope of this paper where we present proof-of-concept measurements. It is also possible to generate contributions to $Q$ from purely electromagnetic coupling from the AF source signal, but we find these to be small compared to the acoustic coupling (see Supplementary Note: 3 for supporting details).

The mixing product signal amplitudes $A(f_1 \pm f_a)$ agree well with each other, and track with the amplitude of $A(f_a)$ (Fig. 3). While the frequency dependence of the dielectric constant of the PZT block can contribute to differences in $A(f_1 + f_a)$ and $A(f_1 - f_a)$, we primarily attribute this difference to interferometer instability over time. The trend in increasing difference between $A(f_a)$ and $A(f_1 \pm f_a)$ as a function of AF source frequency with increasing $f_a$ is likely caused by two effects: (1) different phase and loss constant for the signal at frequency $f_a$ between 0.1 and 10 MHz and 5.9 GHz $\pm f_a$, and (2) variation in the electrostrictive properties of the PZT over the experiment frequency range. In the context of our model in Eq. (2), this corresponds to a $Q$ value that is varying slowly as a function of acoustic frequency for the range of the measurement.

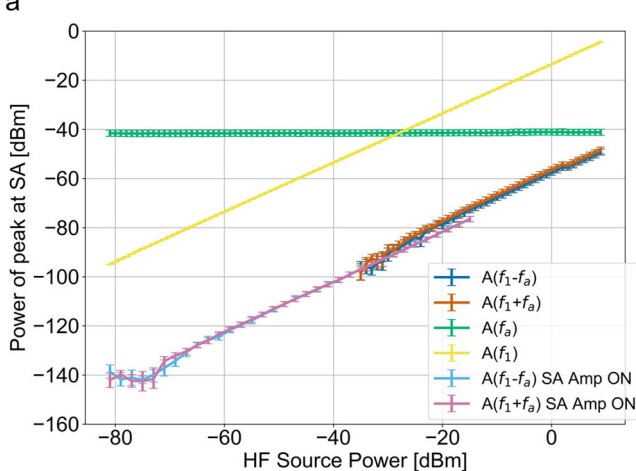

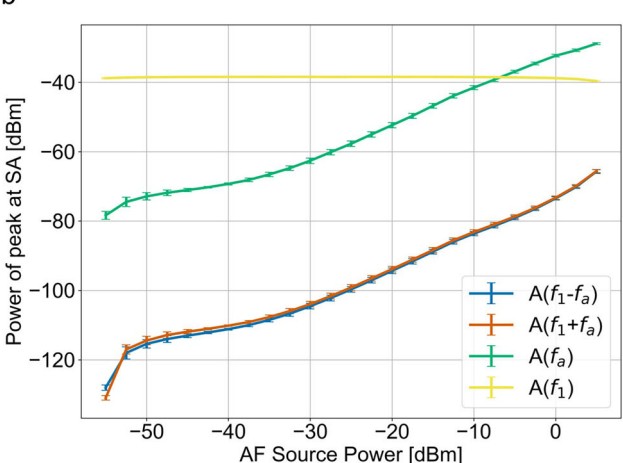

**Fig. 4 | Power dependence of electric-acoustic signal. a** Power of electric-acoustic product mixing peak $A(f_1 \pm f_a)$ and linear signals $A(f_1)$ and $A(f_a)$ as a function of high frequency (HF) source power. Lighter shade of red and blue is a power level for spectrum analyzer (SA) preamplifier turned on, more intense shade is for pre-amplifier turned off. **b** Power of electric-acoustic product mixing $A(f_1 \pm f_a)$ and basic mixing components $A(f_1)$ and $A(f_a)$ as a function of acoustic frequency (AF) transducer source power.

To characterize the power linearity of the sample and the system, we varied the input power of the HF source (Fig. 4a). The measured HF signal power $A(f_1)$ increased linearly with input power and $A(f_a)$ remained constant, as expected. To be able to detect $A(f_1 \pm f_a)$ for lower input power signals, it was necessary to use the built-in preamplifier in the spectrum analyzer (pink and light blue lines in Fig. 4a). For these measurements, we used configuration 2 described in the Interferometer Measurements section above. At higher power levels, we do not use the pre-amplifier since the amplifier goes into saturation and artificially suppresses the signal output. For these measurements, we use configuration 1 described in the Interferometer Measurements section above. In the intermediate power range where both pre-amplified and unamplified signals should be valid, we found good agreement between both measurement approaches, and power-level differences were mostly within their respective uncertainties (see Supplementary Note: 4 for supporting details).

Within the power range of this experimental setup, we observed a linear relationship between the input electrical power $A(f_1)$ signal and the second order mixing products $A(f_1 \pm f_a)$. We characterized the power linearity of the system and sample by varying the input power level of the AF source at 2.2 MHz (Fig. 4b). Here, the dependence of $f_a$ was not as linear as the electrical case, and there were several contributing factors. First, AMP1 is

not as linear as the HF source; AMP1 limits the upper input power to almost 5 dBm, adding 30 dB to the input power level. Small differences of $A(f_1 \pm f_a)$ compared to $A(f_a)$ power are likely caused by changes in temperature influencing the transducer and sample linear acoustic properties. Temperature stability was not an issue for the other experiments we present due to the steady state of acoustic power and air cooling. However, during the acoustic power sweep, we were not able to keep the temperature constant as readily, and it varied over a range of 23–33 °C. We also observed that at higher AF power levels, the peak associated with the electrical signal $f_1$ decreased slightly, which we attribute to temperature drift in the system affecting the linear electrical properties of the sample. For a discussion of the noise floor of the measurement system, see Supplementary Note: 5.

The linear dependence of $A(f_1 \pm f_a)$ on $A(f_1)$ and $A(f_a)$ is a key finding of this measurement technique. This result supports the model that we are measuring a signal exclusively associated with the electrostriction coefficient of the PZT, as proposed in Eq. (2). A nonlinear relationship between the mixing product and the input signals would suggest the contribution of higher order nonlinear materials properties. We conclude that this measurement approach can quantify the electrostriction of PZT, but not higher order nonlinear properties.

## Conclusion
We developed an interferometer-based approach to measuring nonlinear electric-acoustic heterodyning in piezoelectric materials. To demonstrate our approach, we characterized a sample of lead zirconium titanate piezoelectric material and measured the electric-acoustic mixing products for an electrical signal of ~5 GHz and acoustic signals of ~1–10 MHz. We found that the amplitude of this heterodyne signal is linear with electrical power and acoustic power delivered to the sample. We confirmed the electric-acoustic nature of the heterodyne signal by developing a series of diagnostic experiments. This approach can be extended to different sample materials and geometries and can be applied over a wide range of acoustic and electric frequencies. Looking forward, we plan to apply this approach to the study of nonlinear electric-acoustic phenomena in liquids and solids for sensor technologies, biomedical imaging, and chemical characterization.

## Data availability
All the data that support the findings of this study are available in NIST Public Data Repository with the identifier https://doi.org/10.18434/mds2-3828 at "https://datapub.nist.gov/od/id/mds2-3828".

## Code availability
All the code is included with data at the link above with no restrictions.

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

## Acknowledgements

We thank James Booth, Malgorzata Musial and Florian Bergmann for their initial review and improvements to the manuscript.

## Author contributions

T.K. designed and conducted experiments, gathered and analyzed data and wrote the manuscript. R.L. helped with experiments. A.H. and N.O. codesigned theory and experiments. A.S. codesigned theory and experiments, analyzed data, wrote the manuscript and supervised the study. All authors reviewed and approved the manuscript.

## Competing interests

The authors declare no competing interests.
