## [Transparent Peer Review file · Communications Physics]

Measuring Electric-Acoustic Heterodyning in Piezoelectric Materials

Corresponding Author: Dr Angela Stelson

Version 0:

Reviewer comments:

Reviewer #1

(Remarks to the Author)

The paper reports the use of a microwave interferometer to measure the frequency mixing (i.e., heterodyning) due to the solid-state acoustoelectric effect (also called the piezoelectric effect).

The acoustic field and the electric field are simultaneously applied to a piezoelectric crystal (i.e., piezoelectric lead zirconium titanate (PZT)) at different frequencies, resulting in the generation of electric fields at the sum and difference frequencies of the two fields that are measured by the interferometer. The paper describes the principle of an interferometer, the setup including its characterisation, and a PoC validation of the approach.

The technical description of the approach is good. However, I am not sure the paper makes a sufficient methodological advance to justify publication in Communications Physics. The solid-state piezoelectric effect has been extensively investigated over the last decade, albeit more so in converting between acoustic and electric fields. Yet, the authors do not contextualise their approach to earlier methods for measuring piezoelectric heterodyning and explain why their approach is superior. The use of an interferometer to measure weak microwave signals is not novel and in fact has been used already with piezoelectric crystals (e.g., DOI:10.1109/tuffc.2003.1182115).

Reviewer #2

(Remarks to the Author)

This paper presents an experimental approach to measuring nonlinear signal mixing using a PZT material operating at 5 GHz, with a background signal in the low MHz regime. The topic is interesting and relevant; however, the presentation of the material detracts from its impact. In particular, the manuscript includes ancillary commentary that could be streamlined or removed. Additionally, there is a need for more detailed discussion on the nonlinear signal mixing phenomena being measured with the PZT material.

At present, much of the discussion is centered on testing procedures and how they influence the measurement process. While I understand the paper's primary focus is on the experimental methodology, a more substantive discussion of the results—including whether they align with expectations—is necessary for publication.

1. The manuscript includes several instances where detailed information is presented unnecessarily, detracting from the focus of the paper. For example, the discussion of the Gibbs free energy leading to the nonlinear electrostriction constant—while physically correct—does not contribute meaningfully to the experimental approach or the interpretation of the results. Instead of including this derivation, the authors could simply introduce the nonlinear constitutive relation or note that PZT exhibits electrostrictive behavior, with an appropriate reference. These types of extraneous theoretical details dilute the impact of the paper and should be streamlined to maintain focus. Note this is one example

2. The nonlinear mixing results presented appear to be based on rich experimental data, yet the discussion falls short in demonstrating the effectiveness of the measurement approach. The authors should use this opportunity to convince the reader that the experimental procedure is functioning as intended by clearly explaining the observed trends and underlying physics. While most of the discussion is given to potential measurement issues, the discussion lacks a substantive evaluation of the results themselves—how well they align with expectations, and why this measurement technique is particularly effective. Strengthening this section would significantly enhance the paper's impact and credibility.

In summary, the authors should remove ancillary material that does not meaningfully contribute to the core message of the paper. At the same time, they should expand the discussion of the nonlinear mixing results—explaining why the measurements make sense from a physical standpoint and articulating the strengths of their experimental technique. A clearer focus on these aspects will significantly improve the paper's clarity and overall impact.

Reviewer #3

(Remarks to the Author)

The paper describes an experiment where MHz-frequency acoustic waves are mixed with GHz frequency electromagnetic waves resulting in the production of sum and difference frequencies: $f_1 \pm f_a$, where f_1 is the electromagnetic wave frequency and f_a the acoustic frequency. The nonlinear element is in a piezoelectric material. Owing to the relatively small acoustic frequency, the sum and difference frequencies fall very close to f_1 and could be easily lost in the measurement due to the very strong f_1 signal, so the authors use an interferometer arrangement to cancel the f_1 contribution.

The experiment is clearly explained and the results are convincing that the authors have detected mixing of electromagnetic and acoustic signals. I believe the results make a useful contribution to the field and are worthy of publication. However, I do not think the paper in its present form is ready for publication and the authors should consider some changes, see below.

The motivation of this work is not very clear: the common reason for mixing signals like this is to produce an 'intermediate' frequency signal that can be more easily processed (e.g. digitised) or for up-conversion of modulated signals. Here an acoustic signal at around 2 MHz is shifted to a frequency of ~ 6 GHz, which would seem to complicate signal processing when the 2 MHz signal could easily be converted directly in the electric domain using a PZT.

I assume, therefore, the main aim of the work is to demonstrate that it is possible to achieve mixing signals in the electromagnetic and acoustic domain. In this case the authors need to explain more clearly the relationship of their technique to other published works in the area, both those cited in their paper and others, e.g. S Heywood, et al. Sci Rep 6, 30396 (2016). For example, could the author's technique be used to heterodyne a GHz acoustic signal to MHz frequencies or even to DC (homodyne detection) where it can be more easily processed? Could it be applied in the hard to access THz domain? Addressing these points would make a stronger case for the novelty and importance of this work.

An additional point regarding the text on p2 and equations (1) and (2): I don't think a paper for the target audience of CommsPhys needs to include an undergraduate-level explanation of destructive interference.

Version 1:

Reviewer comments:

Reviewer #1

(Remarks to the Author)

The authors addressed the concern that measurement of the piezoelectric effect in piezo materials is well established by elaborating on the difference in measuring the material's nonlinear acoustoelectric frequency mixing originating in the electrostrictive effect vs the linear piezoelectric effect. This is very helpful.

Yet, the authors state that there is a lack of techniques for measuring the frequency mixing in acoustoelectric material ("While these examples illustrate the promise of leveraging nonlinear electric-acoustic materials properties, metrology techniques to quantify these materials properties are lacking."), but did not support this statement. It is unclear what the challenges are in making such measurements. How has frequency mixing been measured to date in materials, and what are the advantages and disadvantages of the proposed interferometer approach? For example, the authors previously reported the use of a vector network analyser-based approach (Lurette, Robert, et al. "Quantifying nonlinear electric-acoustic mixing in lead zirconium titanate with a vector network analyser." Applied Physics Letters 127.2 (2025)). Contextualising the reported work with the measurement challenges and hitherto measurement approaches would help the reader appreciate the methodological advances the paper presents.

Reviewer #2

(Remarks to the Author)

This is my second review of the manuscript, which focuses on measuring the nonlinear mixing between two signals in a piezoelectric material. The authors have addressed the reviewers' concerns, albeit in a minimalistic fashion. While their responses could have included a more thorough discussion of the results, I will defer to the editor and other reviewers to

determine whether the current level of revision is sufficient. From this reviewer's perspective, the changes meet the minimum threshold for acceptance based on my initial comments.

Upon further review—and with the benefit of the other reviewers' insights and the authors' response—it is now apparent that the experimental setup is intended to measure the electrostrictive constants of a piezoelectric. If that is indeed the case, it would have been valuable for the authors to use their measurements and analysis to extract the relevant nonlinear coefficients and include a discussion on accuracy and sensitivity compared to classical methods. Doing so would have provided quantitative results to support their proposed method for measuring nonlinear properties. However, as this point was not raised in my initial review, I will not request a further revision on this basis.

Reviewer #3

(Remarks to the Author)

With the revisions the author's have made to the manuscript, the aim of the work its context and contribution to the field are now much clearer. The authors have addressed all my substantial criticisms and those of the other reviewers, and I am happy to recommend publication in Communications Physics.

Version 2:

Reviewer comments:

Reviewer #1

(Remarks to the Author)

The authors elaborated on the challenges in measuring the nonlinear electric-acoustic frequency mixing (listing low coupling efficiency requiring the application of strong fields and frequency-dependent nature).

They listed a vector network analyser approach that they published earlier this year as the only other alternative measurement approach and suggested that the interferometer approach reported in this manuscript has a lower noise floor, hence higher sensitivity, and is more flexible and lower in cost.

Together, these addenda can help the reader appreciate the methodological advances presented in the paper, thereby addressing the concern raised.

I lack the experience to assess the validity of the advantages stated by the authors. However, I encourage the authors to strengthen the claims they made with evidence, especially since their earlier publication using vector analysis (ref 12) shows a superior -110dBm noise floor (Figure 2), than the interferometer 100dBm (Figure 3) reported here.

Dr. Angela C. Stelson
Guided Wave Electromagnetics Group
National Institute of
Standards and Technology
325 Broadway
Boulder, CO 80305

RE: Decision on Manuscript #COMMSPHYS-25-0845

July 21st, 2025

Dear Editor,

We are submitting our revised manuscript “Measuring Electric-Acoustic Heterodyning in Piezoelectric Materials” for publication in *Communications Physics*. In the enclosed letter, we include the amendments to the manuscript and our response to the referees’ comments and corrections.

Before we begin, we gratefully acknowledge the critical feedback of the referees. Their comments have been extremely useful in making us aware of some points that we were trying to convey yet remained obscure even to well-informed readers. As per the referees' suggestions, we have improved the paper by expanding on several sections, adding clarifications at crucial spots and expanding the description of the materials properties that lead to the signals we measure. We have improved the messaging of the paper to reflect the motivation for sensitive nonlinear materials metrology. Below, we provide each individual comment and the corresponding change to the manuscript. All orange text denotes changes in the manuscript. All original comments are in black, specific questions and requests are enumerated, and their associated responses are in blue text.

In closing, we are excited to have our manuscript join the literature on pivotal advancements in the critical field of nonlinear materials metrology. We thank the reviewers again for their help improving our work.

Thank you very much for your time and consideration.

Sincerely,

Angela Stelson

Microwave Impedance Project Leader
Communications Technology Laboratory
National Institute of Standards and
Technology, Boulder CO.

Original comments from to Reviewer #1:

Summary

The paper reports the use of a microwave interferometer to measure the frequency mixing (i.e., heterodyning) due to the solid-state acoustoelectric effect (also called the piezoelectric effect).

The acoustic field and the electric field are simultaneously applied to a piezoelectric crystal (i.e., piezoelectric lead zirconium titanate (PZT)) at different frequencies, resulting in the generation of electric fields at the sum and difference frequencies of the two fields that are measured by the interferometer. The paper describes the principle of an interferometer, the setup including its characterisation, and a PoC validation of the approach.

Response to Reviewer #1:

The technical description of the approach is good. However, I am not sure the paper makes a sufficient methodological advance to justify publication in Communications Physics. The solid-state piezoelectric effect has been extensively investigated over the last decade, albeit more so in converting between acoustic and electric fields. Yet, the authors do not contextualise their approach to earlier methods for measuring piezoelectric heterodyning and explain why their approach is superior. The use of an interferometer to measure weak microwave signals is not novel and in fact has been used already with piezoelectric crystals (e.g., DOI:10.1109/tuffc.2003.1182115).

We thank the reviewer for this thoughtful and constructive feedback. We agree that the piezoelectric effect and its role in coupling acoustic and electric fields have been extensively studied, particularly in the context of energy conversion and signal transduction. This work focuses on a distinct materials property that is also present in piezoelectric materials: the electrostrictive effect. This nonlinear materials property determines the second-order nonlinear mixing that occurs when a material is subjected to both an electric and an acoustic field. This effect is much smaller than the direct piezoelectric effect and requires advanced techniques to detect the signal associated with this mixing. In response to this feedback, we have restructured the introduction to more clearly highlight the impact of this technique as a measurement of nonlinear materials properties beyond the piezoelectric coefficient. (Manuscript p. 2 pars. 1-3).

In particular, we have added the following sentences to the introduction (p. 1 par. 1):

These applications rely on nonlinear materials physics beyond the piezoelectric effect to create electric-acoustic heterodyned signals.

(p. 1 par. 3):

Inspired by nonlinear electrical measurements and interferometer measurements of the piezoelectric coefficient, we developed an interferometer-based approach to measure

nonlinear electric-acoustic signals. Nonlinear electric-acoustic materials properties beyond the piezoelectric effect are typically small and difficult to detect.

Original comments from Reviewer #2:

This paper presents an experimental approach to measuring nonlinear signal mixing using a PZT material operating at 5 GHz, with a background signal in the low MHz regime. The topic is interesting and relevant; however, the presentation of the material detracts from its impact. In particular, the manuscript includes ancillary commentary that could be streamlined or removed. Additionally, there is a need for more detailed discussion on the nonlinear signal mixing phenomena being measured with the PZT material.

At present, much of the discussion is centered on testing procedures and how they influence the measurement process. While I understand the paper's primary focus is on the experimental methodology, a more substantive discussion of the results—including whether they align with expectations—is necessary for publication.

Response to Reviewer #2:

1. The manuscript includes several instances where detailed information is presented unnecessarily, detracting from the focus of the paper. For example, the discussion of the Gibbs free energy leading to the nonlinear electrostriction constant—while physically correct—does not contribute meaningfully to the experimental approach or the interpretation of the results. Instead of including this derivation, the authors could simply introduce the nonlinear constitutive relation or note that PZT exhibits electrostrictive behavior, with an appropriate reference. These types of extraneous theoretical details dilute the impact of the paper and should be streamlined to maintain focus. Note this is one example... In summary, the authors should remove ancillary material that does not meaningfully contribute to the core message of the paper.

We appreciate this feedback from the reviewer, and have streamlined the paper in several areas:

1. We have removed equations (1) and (2). We have streamlined the text to directly consider the case where the interferometer is operated at different frequencies.
2. We have moved the derivation for the electrostrictive coefficient and its relationship with the measure mixing product to the supplemental information.
3. We have restructured the introduction to emphasize the connection between our measurements and nonlinear materials properties, to minimize the need for derivations of nonlinear materials properties later in the text and provide more context.

2- The nonlinear mixing results presented appear to be based on rich experimental data, yet the discussion falls short in demonstrating the effectiveness of the measurement approach. The authors should use this opportunity to convince the reader that the experimental procedure is functioning as intended by clearly explaining the observed trends and underlying physics. While most of the discussion is given to potential measurement issues, the discussion lacks a substantive evaluation of the results themselves—how well they align with expectations, and why this measurement technique is particularly effective. Strengthening this section would significantly enhance the paper’s impact and credibility... At the same time, they should expand the discussion of the nonlinear mixing results—

a. explaining why the measurements make sense from a physical standpoint and

a. We thank the reviewer for their comment, we have added the following discussion to the description of the interferometer model (p. 5 par. 7):

In this model of the material, we can expect Q to vary smoothly as a function of electrical and acoustic frequency. However, Q should be independent of the amplitude of the electric and acoustic input signals.

we have added the following discussion to the description of the interferometer model (p. 6 par. 2):

While the frequency dependence of the dielectric constant of the PZT block can contribute to differences in $A(f_1 + f_a)$ and $A(f_1 - f_a)$, we primarily attribute this difference to interferometer instability over time... In the context of our model in Eq. 4, this corresponds to a Q value that is varying slowly as a function of acoustic frequency for the range of the measurement.

b. articulating the strengths of their experimental technique. A clearer focus on these aspects will significantly improve the paper’s clarity and overall impact.

b. We thank the reviewer for their comment, we have added a paragraph at the end of the results to summarize the strengths of this technique (p. 6 par. 5):

The linear dependence of $A(f_1 \pm f_a)$ on $A(f_1)$ and $A(f_a)$ is a key finding of this measurement technique. This result supports the model that we are measuring a signal exclusively associated with the electrostriction coefficient of the PZT, as proposed in Eq. 2. A nonlinear relationship between the mixing product and the input signals would suggest the contribution of higher order nonlinear materials properties. We conclude that this measurement approach can quantify the electrostriction of PZT, but not higher order nonlinear properties.

Original comments from Reviewer #3:

The paper describes an experiment where MHz-frequency acoustic waves are mixed with GHz frequency electromagnetic waves resulting in the production of sum and difference frequencies: $f_1 \pm f_a$, where f_1 is the electromagnetic wave frequency and f_a the acoustic frequency. The nonlinear element is a piezoelectric material. Owing to the relatively small acoustic frequency, the sum and difference frequencies fall very close to f_1 and could be easily lost in the measurement due to the very strong f_1 signal, so the authors use an interferometer arrangement to cancel the f_1 contribution.

The experiment is clearly explained and the results are convincing that the authors have detected mixing of electromagnetic and acoustic signals. I believe the results make a useful contribution to the field and are worthy of publication. However, I do not think the paper in its present form is ready for publication and the authors should consider some changes, see below.

Response to Reviewer #3:

1. The motivation of this work is not very clear: the common reason for mixing signals like this is to produce an 'intermediate' frequency signal that can be more easily processed (e.g. digitised) or for up-conversion of modulated signals. Here an acoustic signal at around 2 MHz is shifted to a frequency of ~ 6 GHz, which would seem to complicate signal processing when the 2 MHz signal could easily be converted directly in the electric domain using a PZT.

I assume, therefore, the main aim of the work is to demonstrate that it is possible to achieve mixing signals in the electromagnetic and acoustic domain. In this case the authors need to explain more clearly the relationship of their technique to other published works in the area, both those cited in their paper and others, e.g. S Heywood, et al. Sci Rep 6, 30396 (2016). For example, could the author's technique be used to heterodyne a GHz acoustic signal to MHz frequencies or even to DC (homodyne detection) where it can be more easily processed? Could it be applied in the hard to access THz domain? Addressing these points would make a stronger case for the novelty and importance of this work.

We thank the reviewer for their comment and have adjusted the introduction to better contextualize the work within the field of nonlinear materials measurements. By trying to connect the work with classic heterodyning applications as highlighted by the reviewer, we inadvertently misdirected the reader. This technique is not really designed to create IF signals, or upconvert signals. Rather, the technique is focused on measuring the intrinsic nonlinear electric-acoustic properties of materials, using PZT as a testbed material. These nonlinear properties can be harnessed for a range of applications, as highlighted in the new version of the introduction.

To contrast with work that integrates heterodyning, but relies on the direct piezoelectric effect (e.g. S Heywood, et al. Sci Rep 6, 30396 (2016)), we added the following sentence to the introduction (p.2 par. 1):

These applications rely on nonlinear materials physics beyond the piezoelectric effect to create electric-acoustic heterodyned signals.

2. An additional point regarding the text on p2 and equations (1) and (2): I don't think a paper for the target audience of CommsPhys needs to include an undergraduate-level explanation of destructive interference.

We thank the reviewer for their comment, and we have removed equations (1) and (2). We have streamlined the text to directly consider the case where the interferometer is operated at different frequencies.

Original comments from to Reviewer #1:

Summary

The authors addressed the concern that measurement of the piezoelectric effect in piezo materials is well established by elaborating on the difference in measuring the material's nonlinear acoustoelectric frequency mixing originating in the electrostrictive effect vs the linear piezoelectric effect. This very helpful.

Response to Reviewer #1:

Yet, the authors state that there is a lack of techniques for measuring the frequency mixing in acoustoelectric material ("While these examples illustrate the promise of leveraging nonlinear electric-acoustic materials properties, metrology techniques to quantify these materials properties are lacking."), but did not support this statement. It is unclear what the challenges are in making such measurements.

We have included more context in the introduction to highlight the challenges of nonlinear electric-acoustic materials measurements (p. 1 par. 3):

The biggest challenge to measuring nonlinear electric-acoustic materials properties is that the signals associated with these effects are small, and high power density of both electrical and acoustic signals is typically required [12]. Additionally, electric-acoustic properties are expected to vary as a function of both electric and acoustic frequency, as well as crystal orientation. A robust measurement for these properties should be highly sensitive, extensible to a range of frequencies, and allow for a range of sample orientations and geometries.

How has frequency mixing been measured to date in materials, and what are the advantages and disadvantages of the proposed interferometer approach? For example, the authors previously reported the use of a vector network analyser-based approach (Lurette, Robert, et al. "Quantifying nonlinear electric–acoustic mixing in lead zirconium titanate with a vector network analyser." Applied Physics Letters 127.2 (2025)). Contextualising the reported work with the measurement challenges and hitherto measurement approaches would help the reader appreciate the methodological advances the paper presents.

We have included more context in the introduction to compare the advantages of the interferometer approach we present here compared to the recent work on electric-acoustic measurements referenced by the reviewer (p. 1 par. 4):

The interferometer approach developed here is advantageous compared to other nonlinear electric-acoustic techniques in the literature [12] because it is more sensitive due to the reduction of noise floor with the interferometer... The approach we develop here is flexible with low-cost instruments and can be readily scaled to higher electrical powers.

Original comments from Reviewer #2:

This is my second review of the manuscript, which focuses on measuring the nonlinear mixing between two signals in a piezoelectric material. The authors have addressed the reviewers' concerns, albeit in a minimalistic fashion. While their responses could have included a more thorough discussion of the results, I will defer to the editor and other reviewers to determine whether the current level of revision is sufficient. From this reviewer's perspective, the changes meet the minimum threshold for acceptance based on my initial comments.

Upon further review—and with the benefit of the other reviewers' insights and the authors' response—it is now apparent that the experimental setup is intended to measure the electrostrictive constants of a piezoelectric. If that is indeed the case, it would have been valuable for the authors to use their measurements and analysis to extract the relevant nonlinear coefficients and include a discussion on accuracy and sensitivity compared to classical methods. Doing so would have provided quantitative results to support their proposed method for measuring nonlinear properties. However, as this point was not raised in my initial review, I will not request a further revision on this basis.

Response to Reviewer #2:

We thank the reviewer for their feedback, and we agree that the larger scale goal of the work is to extract the electrostrictive coefficients of the material. We have addressed the next steps required to extract these coefficients in the Results section of the manuscript (p. 6 par. 1):

The electrostrictive coefficient m_{ijkl} can be related to the coefficient Q from Eq. (2) by creating a geometric model of the linear electrical and acoustic properties of the coplanar waveguide device¹². These calibrations are beyond the scope of this paper where we present proof-of-concept measurements.

Original comments from Reviewer #3:

With the revisions the author's have made to the manuscript, the aim of the work its context and contribution to the field are now much clearer. The authors have addressed all my substantial criticisms and those of the other reviewers, and I am happy to recommend publication in Communications Physics.

Response to Reviewer #3:

We thank the reviewer for their constructive feedback in improving this manuscript.

Dr. Angela C. Stelson
Guided Wave Electromagnetics Group
National Institute of
Standards and Technology
325 Broadway
Boulder, CO 80305

RE: Decision on Manuscript #COMMSPHYS-25-0845B

November 5th, 2025

Dear Editor,

We are submitting our revised manuscript “Measuring Electric-Acoustic Heterodyning in Piezoelectric Materials” for publication in *Communications Physics*. In the enclosed letter, we include the amendments to the manuscript and our response to the referees’ comments and corrections.

Before we begin, we gratefully acknowledge the critical feedback of the referees. Their comments have been extremely useful in making us aware of some points that we were trying to convey yet remained obscure even to well-informed readers. As per the referees' suggestions, we have included a table in the supplementary information of the noise floor of the different configurations of the interferometer. These results and the associated discussion explain in detail the advantages of this measurement approach over other techniques. Below, we provide each individual comment and the corresponding change to the manuscript. All orange text denotes changes in the manuscript. All original comments are in black, specific questions and requests are enumerated, and their associated responses are in blue text. Additionally, we have implemented all changes in the Editorial Requests Table.

In closing, we are excited to have our manuscript join the literature on pivotal advancements in the critical field of nonlinear materials metrology. We thank the reviewers again for their help improving our work.

Thank you very much for your time and consideration.

Sincerely,

Angela Stelson

Microwave Impedance Project Leader
Communications Technology Laboratory
National Institute of Standards and
Technology, Boulder CO.

Original comments from to Reviewer #1:

Summary

The authors elaborated on the challenges in measuring the nonlinear electric-acoustic frequency mixing (listing low coupling efficiency requiring the application of strong fields and frequency-dependent nature).

They listed a vector network analyser approach that they published earlier this year as the only other alternative measurement approach and suggested that the interferometer approach reported in this manuscript has a lower noise floor, hence higher sensitivity, and is more flexible and lower in cost.

Together, these addenda can help the reader appreciate the methodological advances presented in the paper, thereby addressing the concern raised.

I lack the experience to assess the validity of the advantages stated by the authors. However, I encourage the authors to strengthen the claims they made with evidence, especially since their earlier publication using vector analysis (ref 12) shows a superior -110dBm noise floor (Figure 2), than the interferometer 100dBm (Figure 3) reported here.

Response:

We have included a reference in the main manuscript (p. 7 par. 1) to a detailed discussion of the noise floor:

For a discussion of the noise floor of the measurement system, see Supplementary Note: 5.

We have included in the supplemental a detailed description of the noise floor in the interferometer measurements under different measurement configurations, and a table that shows the improvement in the noise floor that our measurement technique produces:

Supplementary Note 5: Interferometer Noise Floor

The main purpose of Figure 2. c) of the main manuscript is to show HF and nonlinear mixing signal peaks at $A(f_1)$ and $A(f_1 \pm f_a)$ on the interferometer background characteristics. The noise floor level is combination of number of factors. First is the noise floor level of HF Source which for power used for experiments is around -97 dBm at and $A(f_1 \pm f_a)$, and gets higher closer to $A(f_1)$. That noise floor level can be lowered using interferometer. In theory at the frequency f_{null} where interferometer has its null the cancelation is full and the transmission level is $-\infty$ dBm (see Fig 1. c) yellow line). In practice the measurement setup noise floor is limited by used detector and its settings. To achieve minimal noise floor for used spectrum analyzer it is necessary to use narrow bandpass IF filter setting. Measurements where the data has been collected for Fig 3 and 4 has been made for 2 kHz span, 10 Hz IF filter setting and averaging of 16, that resulted in 3.3 second sweep time with spectrum analyzer minimal possible noise floor level of around -140 dBm while

preamplifier was turned on. That value can be seen as a limit on Fig 4. a) where the preamplifier was used at lowest HF source power. Purpose of Fig 2 c) was to show all three peaks $A(f_1)$ and $A(f_1 \pm f_a)$ therefore we choose span of 5 MHz chosen. If the same spectrum analyzer settings were used, we would need 1 hour and 23.5 second for that measurement. Since Fig 2 c) goal was to show all measured peaks, we chose IF filter bandwidth of 1 kHz which resulted in 23.1 second sweep time and minimal cancelation artificially increased to -110 dBm (see Fig 2. c)). Finally, preamplifier could not be turned off for that figure in a presence of high power $A(f_1)$ signal that could damage spectrum analyzer receiver, which limited noise floor level to -110 dBm.

Below we present a table with noise floor values for two cases. 1) f_1 set as the fundamental frequency provided by the HF source frequency) to be f_{null} (interferometer null), and 2) f_{null} set as the mixing product frequency $f_1 \pm f_a$. In First column we put noise floor level at $A(f_1 \pm f_a)$ for setup without interferometer where the noise floor is set by HF source, second column shows noise floor levels at $A(f_1 \pm f_a)$ with a use of interferometer, and third column shows noise floor at $A(f_1)$ with interferometer. Note that for the 1) case values in columns 2 and 3 are the same while for case 2) values in columns 3 and 4 are the same. It is due to frequency of acoustic transducer being equal to $f_a = 2.2$ MHz. If the f_a frequency would be lower, the nonlinear mixing peaks $A(f_1 \pm f_a)$ would be closer in frequency to the $A(f_1)$ peak, and for the case 1), value in column 2 would be higher as the noise floor level gets higher the closer in frequency it is to $A(f_1)$ while value at the column 3 would be lower as the cancelation of the interferometer gets bigger closer to the f_{null} frequency. As a result, use of interferometer in that experiment allowed us to lower noise floor level caused by nonlinearities of HF source by 43 dB to the level of -140 dBm.

Table S1. Noise floor for different experimental configurations with and without cancellation from the interferometer.

Interferometer null frequency f_{null}	Noise floor at $A(f_1 \pm f_a)$ without cancellation	Noise floor at $A(f_1 \pm f_a)$ with cancellation	Noise floor at $A(f_1)$ with cancellation
$A(f_1)$	~97 dBm	~97 dBm	-140 dBm
$A(f_1 \pm f_a)$	~97 dBm	-140 dBm	~97 dBm